

# Monitoring, cataloguing and weather scenarios of thunderstorm outflows in the Northern Mediterranean

Massimiliano Burlando[1], Shi Zhang[1,2,3], Giovanni Solari[1]

[1]Department of Civil, Chemical and Environmental Engineering (DICCA), University of Genoa, Genoa 16145, Italy
[2]School of Civil Engineering, Beijing Jiaotong University, Beijing 100044, China
[3]Beijing's Key Laboratory of Structural Wind Engineering and Urban Wind Environment, Beijing 100044, China

*Correspondence to*: Massimiliano Burlando (massimiliano.burlando@unige.it)

**Abstract.** High-sampling rate (10 Hz) anemometric measurements of the "Wind, Ports, and Sea" monitoring network in the Northern Tyrrhenian Sea have been analyzed to extract the thunderstorm-related signals and catalogue them into three families
according to the different time-scale of each event, subdivided among 10 minutes, 1 hour and 10 hours long events. Their characteristics in terms of direction of motion and seasonality/daily occurrence have been analysed: it turned out that most of the selected events come from the sea and occur from 12:00 to 00:00 UTC during the winter season. In terms of peak wind speed, the strongest events all belonged to the 10-min family, but no systematic correlation was found between event duration and peaks.

Three events, each one representative of the corresponding class of duration, have been analysed from the meteorological point of view in order to investigate their physical nature. According to this analysis, which was mainly based on satellite images, meteorological fields obtained from GFS analyses related to convection in the atmosphere, and lightning activity, the thunderstorm-related nature of the 10 min and 1 h events was confirmed. The 10 h event turned out to be a synoptic event, related to extra-tropical cyclones activity.

## 1 Introduction

A primary aim of politics, science and technology is to pursue the safety and cost-efficiency of built environment under natural hazards by forecasting catastrophic events, assessing their precursors, evaluating and managing the risk they induce. Wind is the most destructive natural phenomenon - over 70% of the damage and deaths caused by the nature are due to the wind (Tamura and Cao, 2012; Ulbrich et al., 2013) - so its actions are crucial for the safety and cost of structures, infrastructures
and territory. Their evaluation is thus a cornerstone of civil engineering and atmospheric sciences and a societal need.

A climatologic condition in which wind phenomena of different nature coexist - e.g. extra-tropical and tropical cyclones, monsoons, tornadoes, downslope winds and thunderstorms - is referred to as a mixed wind climate (Gomes and Vickery, 1978). Extra-tropical cyclones are the most typical events that strike the mid-latitude areas. Tropical cyclones, monsoons, tornadoes and downslope winds are key features of the wind climate of specific zones. Thunderstorms occur almost
everywhere. The European wind climate and that of many countries at the mid-latitudes are dominated by extra-tropical cyclones (Deroche et al., 2014) and thunderstorms (Letchford et al., 2002).

The polar front theory (Bjerknes and Solberg, 1922) explains and describes the genesis and life cycle of extra-tropical cyclones. They are synoptic phenomena that develop in a few days on a few thousand kilometres (Fig. 1a). Their surface velocity field is characterized by a mean wind profile in equilibrium with an atmospheric boundary layer whose depth is of the order of
magnitude of 1-3 km. Here, within time intervals between 10-min and 1-h, turbulent fluctuations are stationary and Gaussian. Davenport (1961) identified the most intense wind events with the extra-tropical cyclones and introduced a model, based on this hypothesis, to determine the wind loading of structures. After over half century it is still a foundation of wind engineering (Simiu and Scanlan, 1996; Holmes, 2015).





The modern study of thunderstorms started when Byers and Braham (1949) proved that these events are mesoscale phenomena that develop in a few kilometres (Fig. 1b). They consist of convective cells that evolve in about 30-min through three stages in which an updraft of warm air is followed by a downdraft of cold air. Fujita (1985; 1990) showed that the transient downdraft that impinges on the ground produces intense radial outflows (Fig. 2). The whole of these air movements is called downburst

and is referred to as a macroburst or a microburst depending on whether the downdraft diameter is greater or smaller than 4 km, respectively. Radial outflows exhibit non-stationary and non-Gaussian wind speed properties, and a vertical "nose profile" that increases up to about 50-100 m height, then decreases above. These studies gave rise to an extraordinary fervour of research in atmospheric science (Goff, 1976; Wakimoto, 1982; Hjelmfelt, 1988).

In the same period wind engineering realized that the design wind speed and many catastrophic wind events (Fig. 3) that strike

the mid-latitudes areas are often due to thunderstorm outflows (Letchford et al., 2002). Hence, an extensive research arose, dual to the one that took place in atmospheric science, along four main directions (Solari, 2014): 1) wind statistics in mixed climates (Gomes and Vickery, 1978; Kasperski, 2002); 2) monitoring and data analysis (Choi and Hidayat, 2002a; Holmes et al., 2008; Lombardo et al., 2014; Gunter and Schroeder, 2015; Yu et al., 2016); 3) modelling and simulation by wind tunnel tests (Letchford et al., 2002; Mason et al., 2005; Xu and Hangan, 2008; McConville et al., 2009), computational fluid dynamics

(Selvam and Holmes, 1992; Kim and Hangan, 2007; Mason et al., 2010; Vermeire et al., 2011; Zhang et al., 2013; Aboshosha et al., 2015; Karmakar et al., 2017) and analytical methods (Oseguera and Bowles, 1988; Vicroy, 1992; Holmes and Oliver, 2000; Li et al., 2012; Abd-Elaal et al., 2014; Chen and Letchford, 2004a); 4) wind actions on ideal systems (Choi and Hidayat, 2002b; Chen and Letchford, 2004b; Chen, 2008; Kwon and Kareem, 2009; Le and Caracoglia, 2015) and real structures (Darwish et al., 2010; Aboshosha and El Damatty, 2015; Elawady et al., 2017).

Despite this huge amount of research, however, this matter is still dominated by large uncertainties; even more there is not yet a shared model of thunderstorm outflows and their actions on structures like that developed by Davenport (1961) for extra-tropical cyclones. This happens because the complexity of the thunderstorm downbursts makes difficult to establish physically realistic and simple engineering schemes, their short duration and small size make few data available, and a large gap exists between wind engineering and atmospheric science. It follows that the wind loading of structures is still evaluated by the

Davenport's model without any concern for the real nature and the properties of the Aeolian event that causes the loading. This is nonsense because extra-tropical cyclones and thunderstorm outflows are different phenomena that need separate assessments (Solari, 2014).

The research on thunderstorm outflows carried out by the Wind Engineering and Structural Dynamics (WinDyn) Research Group (www.windyn.org) at the University of Genoa takes cue from two European Projects, i.e. "Wind and Ports" (WP)

(2009-2012) (Solari et al., 2012) and "Wind, Ports and Sea" (WPS) (2013-2015) (Repetto et al., 2018), financed by the European Cross-border Cooperation Program "Italy–France Maritime 2007-2013". They handled the problem of the safe management and risk assessment of North Tyrrhenian seaport areas with respect to strong wind conditions through a joint co-operation between the Windyn group - the unique scientific partner in these projects - and the Port Authorities of Genoa, Savona – Vado Ligure, La Spezia, Livorno (Italy) and Bastia – L'Île-Rousse (France). In this framework, a wide in situ wind

monitoring network has been created that is generating an unprecedented amount of high quality wind measurements. In addition to their institutional role to support the activities of Port Authorities, they represent an unlimited source of information to carry out scientific research in several different fields.

The analysis of these data shows the presence of recordings due to wind phenomena of different nature, namely extra-tropical cyclones, thunderstorms outflows and intermediate events (Kasperski, 2002; Zhang et al., 2017). Thus, in order to focus on

the study of intense thunderstorm outflows a semi-automatic procedure was implemented to recognize and extract these phenomena (De Gaetano et al., 2014). This approach is consistent with previous procedures developed and calibrated in order to process a huge amount of data based on few synthetic elements derived from the sole anemometric recordings (Riera and Nanni, 1989; Twisdale and Vickery, 1992; Choi and Tanurdjaja, 2002; Kasperski, 2002; Duranona et al., 2006; Lombardo et

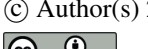


al., 2009), without carrying out systematic and prohibitive meteorological surveys of the weather scenarios out of which they took place. According to this criterion, an extensive set of records labelled as thunderstorms has been gathered and subjected to probabilistic signal analyses aiming at evaluating their main properties relevant to the wind loading of structures (Solari et al., 2015a; Zhang et al., 2017). These properties have formed the base on which two novel methods have been proposed to

determine the structural response to thunderstorm outflows (Solari et al., 2015b; Solari, 2016; Solari et al., 2017).

Despite its inherent advantages and merits, however, this approach suffers the main shortcoming of missing the knowledge of the weather scenarios that occur during events classified as thunderstorms without recognizing their actual meteorological nature. In order to make a first step towards filling this gap, the thunderstorm downburst that occurred on 1 October 2012 over the City of Livorno, Italy was selected as a reference test case (Burlando et al., 2017a). Detailed analyses were carried out of

the wind speed and direction records detected by the WP and WPS network. In parallel, the atmospheric conditions concurrent with this event were studied in great detail by gathering all the meteorological data available in this area, in particular model analyses, standard in-situ measurements (stations and radio-soundings), remote sensing techniques (radar and satellite), proxy data (lightning), and direct observations (from the European Severe Weather Database, Dotzek et al., 2009). This information lead to reconstruct the weather scenario, to classify this event as a wet downburst, to determine its space-time evolution, and

to embed in this framework signal analyses aiming to extract the key parameters for determining the wind loading of structures. From this point of view the above study may become a reference model to carry out comprehensive analyses of the major thunderstorm events recorded by the WP and WPS monitoring network and more in general by any network. This is relevant, however, only repeating such investigations for several events and elaborating the results in a probabilistic framework aiming to construct suitable models of the wind field of thunderstorm outflows truly related to classes of meteorological events, to

recognize clear precursors of these phenomena and their main parameters, to develop statistical analyses of their occurrence in terms of touch-down position, size, motion pattern and background flow. These data are in turn the key information to implement advanced thunderstorm models addressed to hazard and risk analyses for a broad spectrum of applications.

Unfortunately, the framework depicted above may represent a utopian prospect due to the burden of collecting so many data from different sources and performing their joint analysis for several events. Finding a reasonable balance between

expeditionary evaluations based on the sole wind records detected and studies that encapsulate the above information within detailed meteorological surveys is a very difficult and ambitious aim. This paper represents a first step and a pilot attempt in this direction. Section 2 describes the WP and WPS wind monitoring network and dataset. Section 3 illustrates the separation and classification procedure applied in order to gather a rich sub-dataset of records labelled as thunderstorm outflows. Section 4 provides a new procedure to extract and catalogue thunderstorm outflows with reference to their duration and intensity, these

parameters being the key features for evaluating the wind loading of structures. Section 5 furnishes additional elements on the direction, seasonality and hour of daily occurrence of these events. Section 6 introduces a synthetic meteorological survey, coherent with but easier than the general methodology described by Burlando et al. (2017a), of the weather scenarios corresponding to three events preliminarily labelled as "thunderstorms" and characterized by different lifetime-scales. Section 7 summarizes the main conclusions and highlights some prospects for future research.

**2 Monitoring network and dataset**

The wind monitoring and measurement through direct- and remote-sensing techniques is essential for a broad range of scientific disciplines including atmospheric physics, meteorology, climatology, wind, civil and environmental engineering. Besides, the analysis of wind data has an ever more extensive field of application in the interpretation of the wind phenomena that occurred in the past as well as in forecasting the future ones, in climatological analyses and in climate change surveys

(Ortego et al., 2014; Dawkins et al., 2016), in the study of wind hazards, damage and risks to which buildings, infrastructures and human activities in contact with Earth's atmosphere are exposed, until pollutant dispersion and wind energy production.



The WP project gave rise to a wind monitoring network made up of 23 ultrasonic anemometers (yellow circles in Fig. 4) distributed in the Ports of Genoa (2), La Spezia (5), Livorno (5), Savona – Vado Ligure (6) and Bastia (5). WPS, which represents the continuation and development of WP, enhanced and enlarged this network by means of 5 ultrasonic anemometers installed in the Ports of Savona (1), La Spezia (1), Livorno (1) and L'Île-Rousse (2) (orange triangles in Fig. 4),

3 weather stations - each one including an additional ultra-sonic anemometer, a thermometer, a barometer and a hygrometer - in the Ports of Genoa (1), Savona (1), and Livorno (1) (blue diamonds in Fig. 4), and 3 LiDAR (Light Detection And Ranging) wind profilers, again in the Ports of Genoa (1), Savona (1), and Livorno (1) (red squares in Fig. 4), which detect the vertical wind profile at 12 heights from 40 m to 250 m above the ground level (AGL). Other sensors installed autonomously by single Port Authorities are now in the stage of becoming integral parts of the WP and WPS network.

The ultra-sonic anemometric stations consist of bi-axial or three-axial sensors that detect the wind speed and direction with a precision of 0.01 m s$^{-1}$ and 1 degree, respectively. Their sampling rate is 10 Hz with the exception of one sensor installed in Savona, which samples at 1 Hz, and the sensors installed in the Ports of Bastia and L'Île-Rousse, whose sampling rate is 2 Hz. Their position was chosen in order to cover homogeneously the port areas involved in these projects and to register undisturbed wind velocity histories. The instruments are mounted on high-rise towers or on antenna masts at the top of buildings, paying

attention to avoid local effects that may contaminate the quality of measurements. Table 1 illustrates the most important features of the 31 anemometers that currently the WP and WPS network is made up of, $h$ being their height AGL that varies from 10 m to 84 m.

A set of local servers placed in each Port Authority's headquarter receives the data recorded by anemometers and LiDARs in its own port area and elaborates the basic statistics on 10-min of averaging period, namely the mean and peak wind velocities

and the mean wind direction. Each server automatically sends this information to the central server at the University of Genoa, which stores the data in a database through a four-step automatic procedure:

    1)   raw data are systematically checked and validated;

    2)   10-min statistics, including turbulence intensity and gust factor, are evaluated from raw data and stored in the database;

    3)   1-min statistics, including turbulence intensity and gust factor, are evaluated and stored in the database;

4)   an automatic check report is produced and sent to Port Authorities daily.

These results are the main outcome transferred to Port Authorities for their institutional activity. At the same time, they represent the starting point for a broad range of researches carried out by the WinDyn Research Group.

In this regard, starting from the original database, a new one has been created that collects further statistical parameters of the anemometric measurements. In particular, one record for each subsequent $T = 10$-min period is stored that gathers such

parameters into three groups:

    1)   1-s peak wind velocity $\hat{V}$, mean wind velocity $V_{m10}$, mean wind direction $\alpha_{m10}$, gust factor $G_{10} = \hat{V}/V_{m10}$, turbulence intensity $I_{10}$, skewness $\gamma_{10}$, and kurtosis $\kappa_{10}$ in the interval $T$;

    2)   mean wind velocity $V_{m60}$, gust factor $G_{60} = \hat{V}/V_{m60}$, turbulence intensity $I_{60}$, skewness $\gamma_{60}$, and kurtosis $\kappa_{60}$ in the 1-h time interval centered around $T$;

3)   maximum mean wind velocity averaged over 1-min $V_{m1}$ and gust factor $G_1 = \hat{V}/V_{m1}$ in $T$.

This dataset represents the starting point for the separation and classification procedure described in the next section. Similar analyses are currently in progress with regard to LiDAR measurements (Burlando et al., 2017b).

## 3 Separation and classification procedure

A thorough examination of the huge amount of data collected during the WP and WPS Projects reveals that intense wind

events can be separated and classified into three families characterized by different statistical properties (De Gaetano et al., 2014):





(1) stationary Gaussian events with relatively large mean wind velocities and small gust factors; they usually correspond to synoptic neutral atmospheric conditions and are here referred to as extra-tropical cyclones or depressions (Fig. 5);

(2) non-stationary non-Gaussian events with relatively small mean wind velocities, large and quite isolated peaks, and high gust factors; they are here referred to as thunderstorms outflows (Fig. 6);

(3) stationary non-Gaussian events with relatively small mean wind velocities, large and repeated peaks, and moderately high gust factors; they are here referred to as intermediate events (Fig. 7) or gust fronts (Kasperski, 2002). While waiting to carry out a systematic meteorological survey and interpretation of these events, it seems to be reasonable to advance the hypothesis that they are associated to strongly unstable atmospheric conditions, or downslope winds, or recirculating vortices as well.

The separation of intense wind phenomena into homogeneous families is a key topic to interpret the events of engineering interest and to deal with them by models coherent with their physical reality. This is possible by inspecting each event and the weather scenario in which it occurs by merging the analysis of the anemometric recordings with that of the meteorological data detected in the same area and concurrent with the event considered (Burlando et al., 2017a). This operation is clearly not possible when many instruments, many years of measurements, and many different events have to be examined as in this case.

To achieve the separation and classification of different intense wind events as easily and efficiently as possible, De Gaetano et al. (2014) developed a semi-automated procedure applied to the records with peak wind velocity $\hat{V}$ greater than 15 m s$^{-1}$. This choice is coherent with thunderstorm analyses carried out by other authors (Choi, 2000; 2004; Duranona et al., 2006) and with the tradition of evaluating the parameters of synoptic events by collecting all records that satisfy the requirement of neutral atmospheric conditions (Solari and Piccardo, 2001; Solari and Tubino, 2002), these including several phenomena of limited

engineering interest. The alternative approach of restricting analyses to thunderstorm outflows with higher peak values (Geerts, 2001; Lombardo et al., 2014) improves the information related to the phenomena of major engineering interest, but reduces the statistical representativeness of the results.

This semi-automated procedure involves a suitable mix of systematic quantitative controls and qualitative expert judgments. The quantitative controls are based on the comparison between the detected values of the gust factors, $G_{60}$, $G_{10}$, $G_1$, and their

reference values, $G_{60}^0$, $G_{10}^0$, $G_1^0$, evaluated by means of numerical simulations (ESDU, 1993; Burlando et al., 2007; 2013) assuming that intense wind speeds occur in neutral atmospheric conditions during synoptic extra-tropical cyclones with stationary Gaussian properties. Conceptually, a wind event is labeled as an extra-tropical cyclone if the ratios $G_{60}/G_{60}^0$, $G_{10}/G_{10}^0$, $G_1/G_1^0$ are small. Instead, it is labeled as a thunderstorm outflow or an intermediate event when the ratio $G_{10}/G_{10}^0$ is large. In this way, De Gaetano et al. (2014) automatically identified as synoptic extra-tropical cyclones over 99.5% of the

records related to intense wind events. The remaining ones were submitted to qualitative expert judgments. These analyses were based on the wind speed and direction raw data detected by nine anemometers in the period 2011-2012.

Figures 5, 6 and 7 show three typical 1-h long records registered by the anemometer 3 of the Port of La Spezia that correspond to an extra-tropical cyclone, a thunderstorm outflow and an intermediate event, respectively. The pictures (a) and (b) show the time-series of the wind speed and direction raw data, respectively. They also provide their mean values over 1-h, $V_{m60}$ and

$\alpha_{m60}$ (solid lines), and 10-min periods, $V_{m10}$ and $\alpha_{m10}$ (dotted lines), and the 1-s peak wind velocity (red circles), which is obviously smaller than the instantaneous peak. The pictures (c) show the ratios $G_{60}/G_{60}^0$, $G_{10}/G_{10}^0$, $G_1/G_1^0$ over subsequent 10-min periods.

The extra-tropical cyclone record shown in Fig. 5 is characterized by a relatively high mean wind speed ($V_{m10}$= 12.04 m s$^{-1}$, $V_{m60}$= 12.21 m s$^{-1}$) and gust peak ($\hat{V}$= 20.46 m s$^{-1}$). The gust factor ($G_{10}$= 1.70, $G_{60}$= 1.68) is rather high but typical of synoptic

neutral atmospheric conditions. Likewise, the wind speed also the wind direction involves stationary features. The ratios $G_{60}/G_{60}^0$= 0.80, $G_{10}/G_{10}^0$= 0.85 and $G_1/G_1^0$= 0.85 are well below 1.

The thunderstorm outflow record shown in Fig. 6 is characterized by a relatively low mean wind speed ($V_{m10}$= 6.60 m s$^{-1}$, $V_{m60}$= 2.99 m s$^{-1}$), a relatively high gust peak ($\hat{V}$= 19.61 m s$^{-1}$) and a very high gust factor ($G_{10}$= 2.97, $G_{60}$= 6.56). The ratio





$G_{60}/G_{60}^0$= 3.69 exhibits a sudden increase in correspondence of the gust peak whereas the ratios $G_{10}/G_{10}^0$= 1.78 is well above 1 and $G_1/G_1^0$= 0.91 is larger than 0.80. The wind direction changes of almost 180° as usually occurs when a downburst passes over the anemometer (Orwig and Schroeder, 2007).

The intermediate event record shown in Fig. 7 is characterized by a relatively low mean wind speed ($V_{m10}$= 9.71 m s⁻¹, $V_{m60}$= 11.85 m s⁻¹) and a rather high gust peak ($\hat{V}$ = 23.75 m s⁻¹); the gust factor ($G_{10}$= 2.45, $G_{60}$= 2.00) is much greater than the typical values in neutral atmospheric conditions, this being confirmed by the ratios $G_{60}/G_{60}^0$= 1.22, $G_{10}/G_{10}^0$= 1.58 and $G_1/G_1^0$= 1.17. Both the wind speed and the direction exhibit quite regular trends without apparent transient features.

It is worth noting that the records shown in Figs. 5-7 have clear trends that do not imply doubtful decisions. Unfortunately, this ideal condition does not always occur and the extraction of the thunderstorm outflows carried out by De Gaetano et al. (2014) may give rise to controversial choices. On the other hand, though not availing of records detected with high sample rates, Duranona (2015) made a fine selection of the convective events that take place in Uruguay by inspecting 10-min mean and peak wind speeds over a period 10-h long. This study inspired a re-calibration of the procedure applied by De Gaetano et al. (2014) based on 10-min, 1-h and 10-h long records. Joined with the possibility of processing a more extensive set of measurements, this approach leads to a novel extraction and cataloguing procedure described in the next section.

## 4 Thunderstorm extraction and cataloguing

As mentioned in the previous section, the above procedure is firstly extended to the data recorded by 14 ultra-sonic anemometers in the period 2011-2015, including the nine anemometers previously inspected in the period 2011-2012 as well. Secondly, following the method suggested by Duranona (2015), the previous analyses have been improved by performing the qualitative selection of thunderstorm outflows based not only on 10-min and 1-h records as in De Gaetano (2014), but also inspecting the 10-h records centered around the peak wind speed.

Table 2 shows a general framework of the anemometers, periods (yyyy.mm.dd) and data herein examined. NTE = 198 and NTR = 277 are, respectively, the number of events and the number of records labelled as thunderstorms; NTR is always greater than NTE since the same thunderstorm event may be detected simultaneously by more than one anemometer.

It is worth noting that the percentage and the total amount of valid data are quite different at each experimental site. This depends first on the successive installation of sensors, then on the different periods in which measurements were not carried out due to accidents or malfunctions of instruments, these including some cases in which they have not been restored yet; there are also periods in which measurements have been judged not enough reliable to be examined and have been disregarded (Cook, 2014a;b). It is also worth noting that the databases of several anemometers have not been analyzed yet, that the monitoring network continuously produces new data and that new anemometers are progressively added to make the network richer and richer.

The analysis developed here involves not only the assemblage of a more comprehensive and controlled thunderstorm outflow dataset but, even more, a major advance in understanding the time-scale of transient events. Accordingly, it favors the classification and cataloguing of such events based on the duration of the transient peak and on the wind speed itself. This information is crucial to investigate the loading and response of structures.

In this regard, transient records have been separated into three families depending on whether the presence of a ramp-up and the transient peak are clearly detectable in 10-min (Fig. 8), 1-h (Fig. 9) or 10-h (Fig. 10) long records; for sake of simplicity they are referred to as "10-min", "1-h" and "10-h" events. It was found that 50.9 % of the extracted transient records are detectable on 10-min periods, 38.3 % of them can be recognized on 1-h periods whereas only 10.8 % are pointed out by inspecting 10-h records. This aspect reflects on the duration of the ramp-up and has a key engineering role. As demonstrated by Kwon and Kareem (2009), the structural response increases on reducing the length of the impulse related to the passage of the gust front. In Figs. 8, 9 and 10 the pictures (a,b), (c,d) and (e,f) refer to 10-min, 1-h and 10-h long records, respectively,



centered around the gust peak; pictures (a,c,e) and (b,d,f) correspond to wind speed and direction, respectively. In all these diagrams the time variation of the wind direction reflects the time variation of the wind speed, exhibiting a change that may be perceived at the same time-scale in which the ramp-up and the transient peak are perceived. Not all the gathered records, however, have the same property.

The diversity between this approach and typical meteorological surveys is apparent. For instance, according to the Federal Meteorological Handbook No. 1 (NOAA, 2005), ''the beginning of a thunderstorm is to be reported as the earliest time: (1) thunder is heard; (2) lightning is observed at the station when the local noise level is sufficient to prevent hearing thunder; or (3) lightning is detected by an automated sensor''. Conversely, ''the ending of a thunderstorm shall be reported as 15 minutes after the last occurrence of any of the above criteria''. The gathering of high quality wind speed records makes it possible to

introduce a diverse duration criterion that represents a key advance for hazard analyses, wind loading modelling, structural response and sensitivity to thunderstorm events.

In addition to the above classification criterion, transient records are separated into four groups as a function of the peak wind speed (in m s$^{-1}$), namely $15 \leq \hat{V} < 20$, $20 \leq \hat{V} < 25$, $25 \leq \hat{V} < 30$, and $30 \leq \hat{V} < 35$. This classification is aimed at recognizing the existence of any correlation between the duration of the gust front and its wind speed; of course, short-duration events with

high wind speed are the most relevant hazard from a structural viewpoint. Figure 11 shows the example of a thunderstorm record for each wind speed class. Table 3 shows the results of intersecting the two classification criteria based on the duration and the wind speed pointing out, at least for the available data, no systematic correlation between the duration of the most intense part of the record and the peak wind speed. It is worth noting, however, that the four events whose peak wind speed exceeds 30 m s$^{-1}$ are characterized by a limited duration. On the other hand, the 21 events whose peak wind speed exceeds

25 m s$^{-1}$ are quite uniformly distributed over different durations.

**5 Thunderstorm direction, seasonality, and hour of daily occurrence**

As shown by Figs. 5-10, the definition of the thunderstorm outflow direction is quite controversial. A thunderstorm cell may be classified as stationary or non-stationary depending on whether it is endowed with a translational motion. In the case of a stationary event the direction of the thunderstorm outflow, of radial nature, strictly depends on the position of the axis of the

downdraft, usually assumed as vertical, with regard to the position of the sensor. In the case of a non-stationary event, the velocity and direction detected by the sensor is the vector composition of the velocity and direction of the thunderstorm cell, dealt with as stationary, and its translational components (Holmes and Oliver, 2000). The situation becomes more complex in the frequent case in which a thunderstorm cell is embedded in a background larger-scale boundary layer flow field, usually of synoptic type, or ever more in the case in which multiple downdraft are generated by single or multiple thunderstorm cells. In

principle, all these situations may be treated by vector compositions of the velocity and direction of component flows; in reality, no proof exists that this approach is physically and mathematically suitable.

Lombardo et al. (2009) identified the thunderstorm outflow direction with its average value in a 5-s period centered around the peak wind speed. Solari et al. (2014) defined it as the average value in a 30-s period centered around the gust peak. In this paper, the period in which the wind direction is averaged is increased to the 1-min centered around the peak.

Figure 12 shows the distribution of the thunderstorm records reported in Table 2 with reference to the day of occurrence shown on the polar radius (1 relates to 1 January) and to the wind direction (0° refers to the north); the background of every diagram is the map of the port for roughly illustrating the relationship between the directions of intense outflows and geographic conditions. Most of the events (68 %) occurs in the months between September and January. Besides, most of the events (78 %) is characterized by wind directions coming from the sea.

In the Port of Savona only one year of reliable data has been gathered, so the number of thunderstorm records is small and no definite trend is identified. It is possible, however, that the spread of these results may be due not only to the scarcity of data



but also to the fact that the Port of Savona comprehends two different port areas: the one of Savona itself and that of Vado Ligure; though these areas are rather close, they have indeed peculiar different features. Also, the presence of intense downslope winds in Vado Ligure area (Burlando et al., 2017c) may contribute to increase the spread.

Figure 13 shows the distribution of thunderstorm records (NTR) corresponding to the three families defined before, namely 10-min, 1-h and 10-h events, reported in Table 2 with respect to the day time. Overall, for the area under study, 20 % of the thunderstorms occur between 00:00 and 06:00 UTC, 24 % occur in the morning between 06:00-12:00 UTC, 30 % occur in the afternoon from 12:00 to 18:00 UTC and 26 % occur in the evening from 18:00 until 00:00 UTC. Therefore, at least in the monitored area, it seems that thunderstorm events are slightly more likely to occur in the warmer hours of the day, possibly because of the role of solar heating in triggering thermals from the earth's surface. The percentage of 1-h events grows in the afternoon with respect to the 10-min events, as well.

It is worth noting, however, that the occurrence of thunderstorms during night hours is an open topic that needs further study. These phenomena are most likely to form when the temperature of the air decreases with height pretty rapidly, i.e. when it is hot at the ground and cold aloft. Thunderstorms that form at night occur in the absence of heating at the ground by the sun, so that they are due to different forcing mechanisms.

## 6 Short meteorological survey and weather scenarios

As previously noted, the procedure depicted by Burlando et al. (2017a) may represent a reference model to carry out comprehensive analyses of the major thunderstorm events. However, its burden is so high that it makes this procedure realistically unusable for systematic analyses of historical series of such events. Hence, the need arises, or at least the objective, to develop an faster approach that integrates the data provided by an anemometric network such as the WP and WPS ones, with few essential meteorological information that may qualify, albeit preliminarily, the transient intense wind events detected by the network.

In this spirit, making treasure of the experience matured during the detailed analysis of several downbursts, in particular the one that stroke Livorno on 1 October 2012 (Burlando et al., 2017a), this section describes simplified meteorological surveys and preliminary reconstructions of the weather scenarios that occurred during the three events presented in Section 4, Figs. 8-10, and already described from the signal analysis viewpoint. In the following three subsections, the meteorological conditions that brought about all these intense wind events, whose characteristic lifetime-scales is 10-min, 1-h, and 10-h, are briefly reported one by one. The analysis is performed at two spatial scales: the meteorological conditions at the synoptic-scale are firstly inspected in order to evaluate the pattern of cyclones, anticyclones, and fronts that determined instability and cloudiness in the atmosphere; the phenomena possibly occurring at the meso-scale that developed over the areas of interest are then investigated in order to understand the specific convective structures present during the events under consideration. The whole analysis is based primarily on the following data: the Global Forecast System (GFS) analyses, obtained from the National Center for Environmental Prediction (NCEP) through the National Centers for Environmental Information database; the cloud top height, obtained from the cloud analysis performed by Eumetsat (EUMETSAT, 2013; Derrien et al., 2013) based on infrared measurements collected by SEVIRI (Spinning Enhanced Visible & Infrared Imager) on board Meteosat Second Generation (MSG) satellites; the lightning strikes, obtained from the Blitzortung database.

GFS analyses are available worldwide on a 0.5° by 0.5° geographical grid with 6-hour time step; these spatial and temporal resolutions are suitable to evaluate the movement of the large-scale structures that determine the evolution of the weather conditions. The satellite measurements of Meteosat 10, which is the one used here, are available every 15 minutes as a full disk imagery centred at 0 degrees of longitude and latitude, with a spatial resolution over Europe of a few kilometres. The spatial and temporal resolution of such data allows to distinguish the convective structures at the meso-scale and their evolution in time with sufficient accuracy also in case of thunderstorms, whose typical scales are of the order of 10 km and 1 h. Finally,

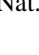


the lightning strikes are used to confirm whether the convective activity can be associated to cumulonimbus clouds, which are typical of thunderstorms, or not.

The meteorological analysis based on such data cannot provide precise information concerning the very small-scale structure of the thunderstorm considered, like the shape of its gust front or the area covered by heavy precipitations. This information can be obtained using higher resolution measurements like radar imagery or finer monitoring networks (Burlando et al., 2017a), and possibly high resolution numerical simulations (Lompar et al., 2017). However, as shown in the next sections, the meteorological analysis described above may be sufficient to state the nature, i.e. convective or synoptic, of the transient intense wind events investigated, which is the main focus of the present paper.

### 6.1 Weather analysis of the 10-minute event on 25 October 2011 in La Spezia

On 25 October 2011, the exceptionally large Anticyclone Ulla (names given by the Institute of Meteorology of the Freie Universität Berlin, Germany) located over Eastern Europe, with pressure maxima greater than 1030 hPa, determined the blocking of the zonal shift of Cyclone Meeno, which remained approximately stationary to the west of Ireland, extending its low-pressure minimum of 985 hPa northward to Iceland, as shown in Fig. 14a. The cold front of Cyclone Meeno, which extended meridionally to northern Africa passing over the Alps, was slightly moving westward over the Mediterranean during the day. The warm sector uplift ahead of Meeno's cold front due to the south-easterly winds forced over southern Italy and the Adriatic Sea by the extension of the influence of Ulla to the Balkans Region, determined a wide area of cloudiness all over northern and central Italy, shown in Fig. 14b. The distribution of cloud top heights in this figure shows that the prevailing southerly flow of warm and humid air from the Mediterranean favoured the development of strong instability associated to deep convection, which in turn determined locally very intense thunderstorm events.

The advection and uplift of moist air from the southern quadrants in the lee of the Alps is confirmed by values of relative humidity higher than 100 % over great part of the Padan Plain, the eastern Liguria and southward along a convergence line which corresponds approximately to the frontal zone beneath. According to the GFS analysis, Fig. 15a shows that saturated air conditions, i.e. RH values (shaded contours) equal to 100%, were occurring exactly above La Spezia, at 18:00 UTC. Moreover, the values of mean storm motion in the 0-6000 m AGL in the same area were between 10 and 15 m s$^{-1}$ from south or southwest. According to satellite measurements, Fig. 15b shows that a deep convective cloud with the top at about 12000 m above the sea level (ASL) occurred at around 15:45 UTC over La Spezia, indeed. However, during the morning of 25 October, a series of thunderstorms developed slightly to the northwest of La Spezia because of the orographic forcing determined by the south-eastern flow, as measured by the anemometers in the Port of La Spezia (see Fig. 8e,f). In that area, many meteorological stations of the Liguria Region (e.g. Monterosso, Serò di Zignago, Levanto S. Gottardo, Brugnato) reported precipitation rates larger than 50 and even 100 mm h$^{-1}$ from 9:00 UTC to 15:00 UTC. The thunderstorm over La Spezia at 15:40 UTC was the last of this series, just before the wind shifted to the north when the cold front overcame the Western Alps. The sudden change in wind speed and direction reported in Fig. 8 represent the transition between these higher-to-lower wind speed regimes. It is worth noting that, as reported in Zhang et al. (2017), the transition between this kind of wind regimes is usually slower, so that these events are often classified as 1-h events. In the present case, the dynamics of the transition is particularly fast so that the event is classified as a 10-min one.

As far as the cloud-ground strikes measured by the Blitzortung network are concerned, this convective cell showed only a moderate lightning activity, which confirms the convective nature of this phenomenon. The lightning occurrence on 25 October 2011, shown in Fig. 16, in the interval from 15:25 to 15:55 UTC (half an hour centred with respect to the maximum wind speed recorded during this event) was 164 strikes, with a slightly increasing frequency during the first 25 minutes.



### 6.2 Weather analysis of the 1-hour event on 4 October 2015 in Livorno

Figure 17 (panels a-b) depicts the synoptic condition over Europe on 4 October 2015, showing the position of cyclones and anticyclones (a) and the cloud cover (b) at 06:00 UTC. The meteorological situation over Europe was dominated by the presence of the anticyclone Netti, with its high-pressure maximum of 1025 hPa situated over Ukraine and Russia, indicating a

blocking condition (Rex, 1950). The cyclone Quirin, which was born on 2 October to the north of the Pyrenees, where a strong thermal contrast between a colder maritime Atlantic air mass to the north and the much warmer and moist tropical air to the south occurred, had slightly moved its low-pressure minimum north-eastward over the northern France on 3 October and Belgium on 4 October. According to GFS analyses, at 06:00 UTC the tropopause height showed an abrupt discontinuity to the north of the Alps with a roughly meridional gradient ranging from a maximum 14214 m to a minimum 9119 m, denoting the

existence of a frontal zone beneath consisting of a warm core of tropical air southward (red contour reported in Fig. 17a) and a colder core of Atlantic air northward (blue contour in Fig. 17a). Cyclone Quirin, which was relatively small but very active because of the strong thermal gradients between Atlantic and tropical air masses, led to intense local thunderstorms and heavy precipitations in the southern France and northern Italy during the night between 3 and 4 October. The distribution of cloud top heights shows clearly the presence of the cyclone Quirin, with its surface low-pressure minimum located over Belgium at

06:00 UTC on 4 October (Fig. 17b) and the occluded front extending southward to the Adriatic Sea.

In the lee side of the western Alps, the presence of a cold front extending from the Gulf of Genoa to the aforementioned occluded front in the northern Adriatic Sea is revealed by high relative humidity values in the lower troposphere due to the forced lift of the warmer and humid air over the Ligurian and Tyrrhenian Sea by the colder air flow from the north-western quadrant. Figure 18a shows that RH values (shaded contours) close to saturation that occurred along a narrow band in the

northern Tyrrhenian Sea, according to the GFS analysis at 06:00 UTC. Vectors in Fig. 18a, which represent the mean storm motion in the 0-6000 m AGL, show that a storm developing along the frontal boundary would eventually move from the sea towards the coast of Tuscany at approximately 5 m s$^{-1}$ or less. This is the case of the thunderstorm shown in Fig. 9 that developed on 4 October early in the morning off the coast of Livorno. The sequence of three satellite images reported in Fig. 18b-d resembles the development of a single-cell thunderstorm caused by a cumulonimbus cloud that started developing

its cumulus stage around 04:30 UTC (Fig. 18b) over Livorno City, then reached the mature stage with a cloud top height of about 12000 m ASL at 05:15 UTC (Fig. 18c), and finally gradually dissipated while moving slowly farther inland (Fig. 18d). Because of the low storm advection from sea to land, the anemometric signals recorded in the Port of Livorno have caught the whole three stages of the thunderstorm evolution on a time scale, i.e. 1 hour, which is approximately coincident to the typical order of magnitude of a single-cell thunderstorm life-cycle.

The intense convective activity that occurred in the surroundings of Livorno City because of this thunderstorm is confirmed by the great number of lightning strikes registered by the Blitzortung network, as reported in Fig. 19. The lightning occurrence from 05:00 to 05:30 UTC was almost 1500 strikes, quite regularly distributed during the 30 minutes (see the frequency histogram in the bottom right corner). Note that over Livorno the whitish symbols, which correspond to times closer to 05:30 UTC, are slightly shifted eastward with respect to the reddish ones, according to the slow thunderstorm advection from

sea to land.

### 6.3 Weather analysis of the 10-hour event on 21 November 2013 in Genoa

The low-pressure system known as Quentin was born on 18 November 2013 in the Baffin Bay, between Canada and Greenland, and moved zonally to the north of England. During the 20 November, it moved south-eastward under the influence of anticyclone Susanne I, located at mid-latitudes over the Atlantic Ocean, and approached Belgium on 21 November at about

00:00 UTC. Then, it moved farther to the south and determined a low-pressure minimum of 995 hPa in the lee side of the Alps that was over the Ligurian Sea at 12:00 UTC, as shown in Fig. 20a. The cold core aloft of Quentin, i.e. tropopause heights (shaded contours) below 8000 m, over France determined strong atmospheric instability and caused high precipitation rates in





Belgium, Holland and France while advecting meridionally during the day. In the lee of the Alps, however, the south-eastward motion of the cold front of Quentin did not induce very deep convection over Liguria, as demonstrated by the relatively low values of the cloud top shown in Fig. 20b, which are between 8000 and 10000 m ASL.

The position of the cold front at the surface in the lee of the Alps at 12:00 UTC is indicated by the high-RH values that extends as an arc-shaped band from the eastern Liguria to the west of Sardinia Island, shown in Fig. 21a. On 21 November in the morning, the front had just passed over the Alps and the secondary pressure minimum aloft determined the wind rotation from north to southwest over the Ligurian Sea. The mean storm motion (vectors in Fig. 21a), which in this case corresponds approximately to the mean flow in the lower half of the troposphere as the directional wind shear from 0 to 6000 m ASL was rather low, was about 20-25 m s$^{-1}$ from southwest, indeed. The strong forcing aloft was likely the main reason for the relatively sudden increase of wind speed recorded by anemometer 2 of the Port of Genoa (Fig. 10) rather than some deep convective phenomenon that did not seem to occur according to the top height of clouds obtained from satellite data (Fig. 21b). This is also confirmed by means of the Blitzortung network that didn't record any strike in this area in between 06:00 and 14:00 UTC.

## 7 Conclusions and perspectives

The wind monitoring network realized for the European Projects "Wind and Ports" and "Wind, Ports and Sea" is an inexhaustible source of measurements that highlight the speed and frequency of transient events, likely of convective nature, often disregarded, especially in the past, from classical wind engineering. It is now recognised that they are crucial with respect to hazard assessments aimed at the safety of construction, infrastructure and territory, but they are often still undistinguished from extra-tropical cyclones in most cataloguing procedures (Stucki et al., 2014).

In the first part (Sections 2-5) this paper provides a description of the main properties of the anemometric network and the database generated by it. It then illustrates the procedure used to separate the records associated with different wind phenomena based on information - stationary and Gaussian features - typical of signal analysis but lacking of meteorological contents. Thanks to this method, events labelled as thunderstorm outflows are selected and classified into three families according to the time-scale - 10-min, 1-h, 10-h - over which the transient part of the wind speed develops. In addition, analyses related to speed, direction, seasonality and hour of daily occurrence are also presented.

The second part of this paper takes cue from a detailed meteorological survey of the wet downburst that occurred on 1 October 2012 over the City of Livorno (Burlando et al., 2017a). Since the burden of this approach prevents its realistic application within systematic analyses of historical series of similar phenomena, an expeditionary procedure is codified and proposed herein to integrate the anemometric records with few essential meteorological features that at least qualify the convective or synoptic nature of different detected events. This procedure has been applied to three sample events referred to as 10-min, 1-h and 10-h long records.

The analysis of the two shorter events have confirmed their convective nature. These thunderstorms, however, have different triggering mechanisms. The 10-min event was determined by the mechanical lift of maritime air exerted by the orography; its transition between higher-to-lower wind speed regimes corresponds to the passage of the cold front. The 1-h event was most probably brought about by the mechanical lift due to the cold front southeastward movement. These remarks may provide preliminary motivations to the different temporal scales of fast transient events and stimulate research towards the comprehension of this delicate issue.

Conversely, the longer event, i.e. the 10-h one, turned out to be a synoptic phenomenon, endowed with a rapid evolution, initially misclassified as a potential thunderstorm, and it should be likely catalogued as an extra-tropical cyclone-related windstorm instead (Roberts et al., 2014). Even if this result cannot be generalized to the whole family of 10-h intense wind events, it raises the question whether in some particular cases these phenomena can really have a convective genesis. This question remains open and will deserve further and more systematic investigations in the future.



The meteorological analysis of the three events considered here may result helpful also clarifying the relation between the shape of anemometric signals, discussed by Zhang et al. (2017), and the underlying meteorological phenomena. It is worth noting, however, that each particular shape could be determined in principle by more than one phenomenon, especially if different locations are considered. In perspective, therefore, this analysis should become more systematic and should be
repeated for different databases of recordings taken at different latitudes and in geographical contexts different from the coastal area considered here.

## Acknowledgements

This research is funded by European Research Council (ERC) under the European Union's Horizon 2020 research and innovation program (grant agreement No. 741273) for the project THUNDERR - Detection, simulation, modelling and loading
of thunderstorm outflows to design wind-safer and cost-efficient structures – supported by an Advanced Grant (AdG) 2016. It is also funded by "Compagnia di San Paolo" for the Project "Wind monitoring, simulation and forecasting for the smart management and safety of port, urban and territorial systems" (grant number 2015.0333, ID ROL: 9820), by Italian Ministry of Instruction and Scientific Research (PRIN 2015), with regard to the Project "Identification and diagnostic of complex structural systems" (grant No. 2015TTJN95), and by 111 Project "Innovation on mitigating wind-induced disaster of
infrastructures sensitive to wind" supported by the Ministry of Education, China, at the Beijing Jiaotong University. The data exploited for this research has been recorded by the monitoring network realized in the course of the European Projects "Winds and Ports" and "Wind, Ports and Sea", funded by European Territorial Cooperation Objective, Cross-border program Italy-France Maritime 2007-2013. Satellite images are based on level 1 data recorded by SEVIRI instrument on board Meteosat Second Generation satellites, operated by EUMETSAT.

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





**Table 1:** Main properties of the WP and WPS monitoring network.

| Port | Anem. No. | Period of Measurement | $h$ (m) | Type | Sampling rate (Hz) |
|---|---|---|---|---|---|
| Savona and Vado | 0 | | 84 | | |
| | 1 | | 33.2 | | |
| | 2 | 2011.03.30-now | 12.5 | tri-axial | 10 |
| | 3 | | 28 | | |
| | 4 | | 32.7 | | |
| | 5 | | 44.6 | | |
| | 6 | 2014.04.05-now | 10 | bi-axial | 1 |
| | 7 | 2015.07.31-now | 35 | | 10 |
| Genoa | 1 | 2011.03.30-2013.05.07 | 61.4 | | |
| | 2 | 2010.10.12-2015.05.31 | 13.3 | bi-axial | 10 |
| | 3 | 2015.04.16-now | 32 | | |
| La Spezia | 1 | 2010.10.29-now | 15.5 | | |
| | 2 | | 13 | | |
| | 3 | 2011.02.04-now | 10 | bi-axial | 10 |
| | 4 | 2011.04.14-now | 11 | | |
| | 5 | 2012.09.06-now | 10 | | |
| | 6 | 2015.01.23-now | 16 | | |
| Livorno | 1 | 2010.09.16-now | 20 | | |
| | 2 | | 20 | | |
| | 3 | 2010.09.16-2015.03.21 | 20 | tri-axial | |
| | 4 | 2010.09.16-now | 20 | | 10 |
| | 5 | 2010.09.16-2014.08.25 | 75 | | |
| | 6 | 2015.07.25-now | 12 | bi-axial | |
| | 7 | | 23.8 | | |
| Bastia | 1 | | 10 | | |
| | 2 | | 10 | | |
| | 3 | 2011.11.17-now | 13 | bi-axial | 2 |
| | 4 | | 10 | | |
| | 5 | | 10 | | |
| L'Île-Rousse | 1 | 2015.06.03-now | 10 | bi-axial | 2 |
| | 2 | 2015.06.08-now | 10 | | |





**Table 2:** Number of thunderstorm events and records examined.

| Port | Anem. No. | Period of Analysis | Valid data | NTE | NTR | 10min | 1h | 10h |
|------|-----------|--------------------|-----------|-----|-----|-------|-----|-----|
| Genoa | 1 | 2011.03.30-2013.04.01 | 59% | 41 | 9 | 5 | 4 | 0 |
| | 2 | 2010.10.12-2015.05.31 | 56% | | 34 | 11 | 18 | 5 |
| Livorno | 1 | 2010.10.01-2015.12.12 | 87% | 84 | 40 | 19 | 14 | 7 |
| | 2 | 2010.10.01-2015.12.12 | 67% | | 20 | 8 | 9 | 3 |
| | 3 | 2010.10.01-2015.03.21 | 74% | | 28 | 15 | 9 | 4 |
| | 4 | 2010.10.01-2015.12.12 | 60% | | 39 | 18 | 19 | 2 |
| | 5 | 2010.10.01-2014.08.25 | 69% | | 16 | 6 | 8 | 2 |
| Savona – Vado Ligure | 1 | 2014.12.01-2016.01.31 | 87% | 23 | 5 | 3 | 2 | 0 |
| | 2 | 2014.12.01-2016.01.31 | 72% | | 3 | 1 | 2 | 0 |
| | 3 | 2014.12.01-2016.01.31 | 83% | | 7 | 6 | 0 | 1 |
| | 4 | 2014.12.01-2016.01.31 | 86% | | 10 | 7 | 2 | 1 |
| | 5 | 2014.12.01-2016.01.31 | 87% | | 4 | 2 | 1 | 1 |
| La Spezia | 2 | 2010.10.29-2015.12.31 | 88% | 50 | 20 | 12 | 8 | 0 |
| | 3 | 2011.02.05-2015.12.18 | 89% | | 42 | 28 | 10 | 4 |
| Total | 14 | - | - | 198 | 277 | 141 | 106 | 30 |
| Percent | - | - | - | - | 100% | 50.9% | 38.3% | 10.8% |



**Table 3:** Classes of membership of the peak wind speed of the thunderstorm outflows.

| Duration | $\hat{V}$ (m/s) | | | | |
| --- | --- | --- | --- | --- | --- |
| | 15-20 | 20-25 | 25-30 | 30-35 | All NTR |
| 10-min | 88 (62%) | 40 (28%) | 9 (6%) | 4 (3%) | 141 |
| 1-h | 78 (74%) | 22 (21%) | 6 (6%) | 0 (-) | 106 |
| 10-h | 20 (67%) | 8 (27%) | 2 (7%) | 0 (-) | 30 |
| All NTR | 186 | 70 | 17 | 4 | 277 |



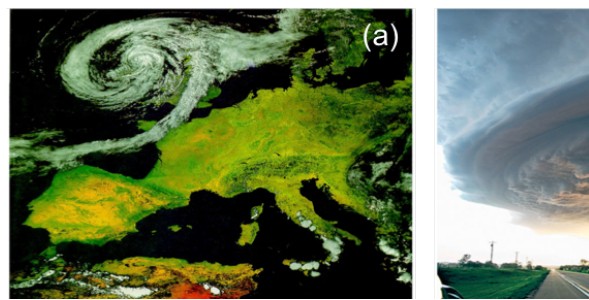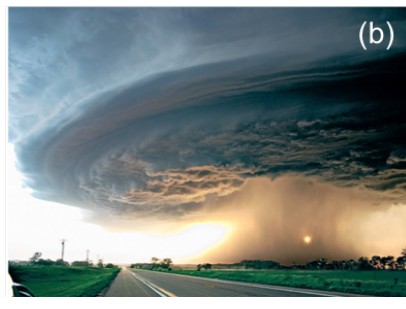

**Figure 1: Synoptic extra-tropical cyclone (a) and mesoscale thunderstorm downburst (b).**





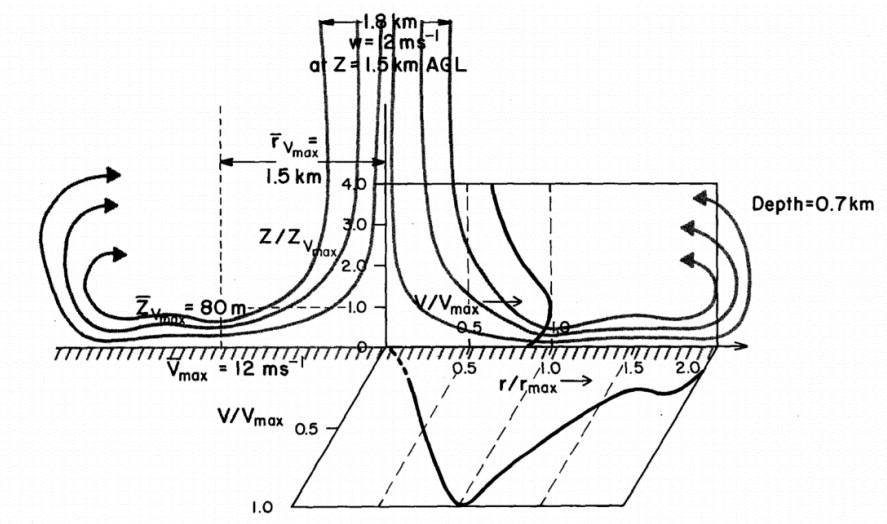

**Figure 2: Scheme of a thunderstorm downburst and nose velocity profile in the radial outflow (After Hjemfelt 1988).**




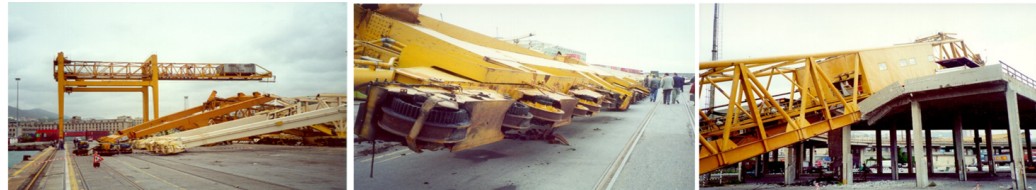

Figure 3: Damages caused by the thunderstorm downburst that occurred in the Port of Genoa (Italy) on 31 August 1994.





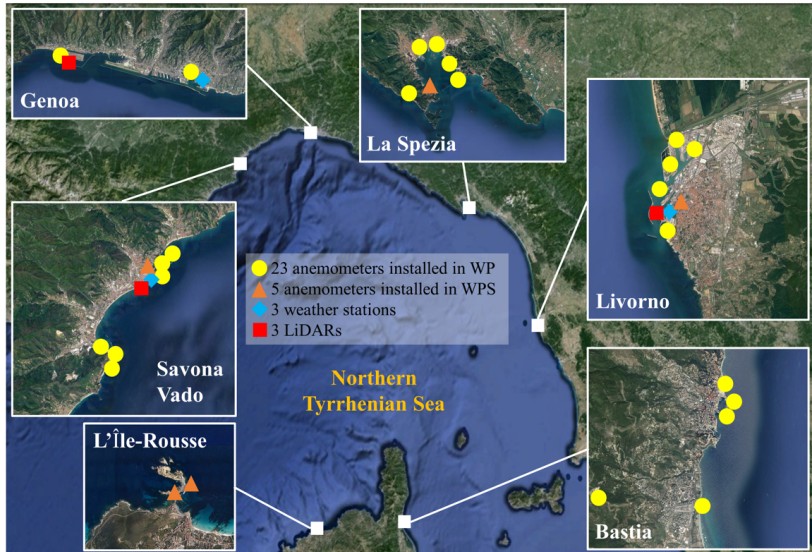

Figure 4: WP and WPS anemometric monitoring network.

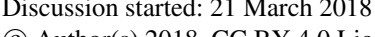



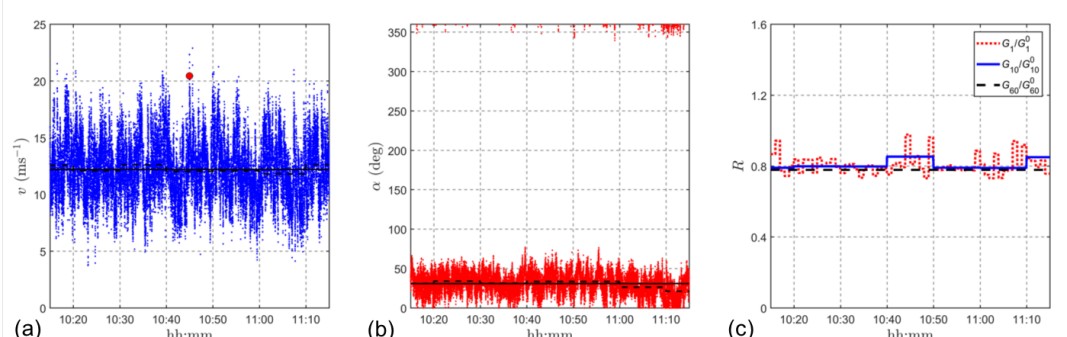

**Figure 5: Extra-tropical cyclone recorded on 1 December 2013 by the anemometer 3 of the Port of La Spezia: (a) 1-h wind speed time-series; (b) wind direction time-series; (c) ratio of gust factors.**





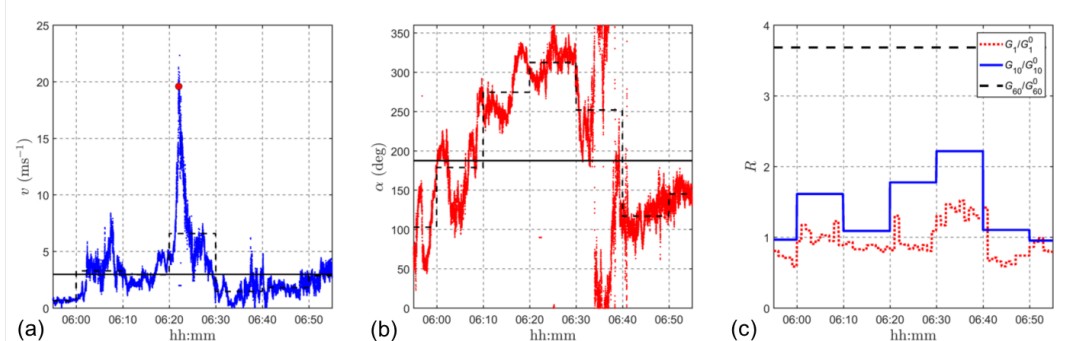

**Figure 6: Thunderstorm outflow recorded on 25 June 2014 by the anemometer 3 of the Port of La Spezia: (a) 1-h wind speed time-series; (b) wind direction time-series; (c) ratio of gust factors.**




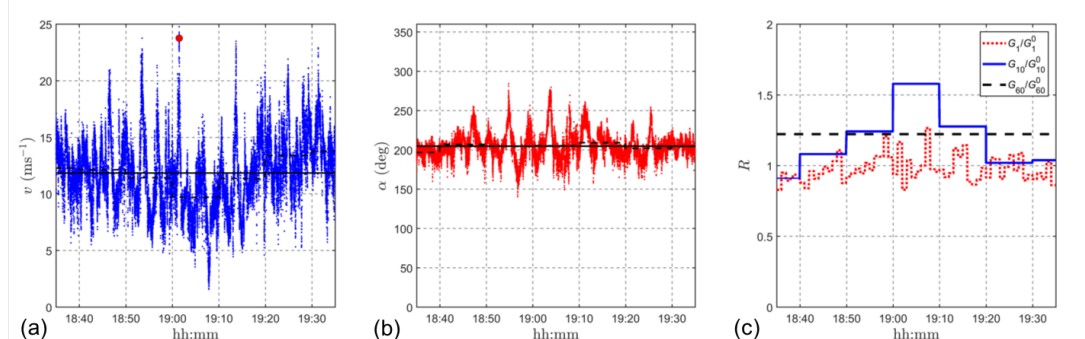

**Figure 7: Intermediate event recorded on 10 October 2013 by the anemometer 3 of the Port of La Spezia: (a) 1-h wind speed time-series; (b) wind direction time-series; (c) ratio of gust factors.**




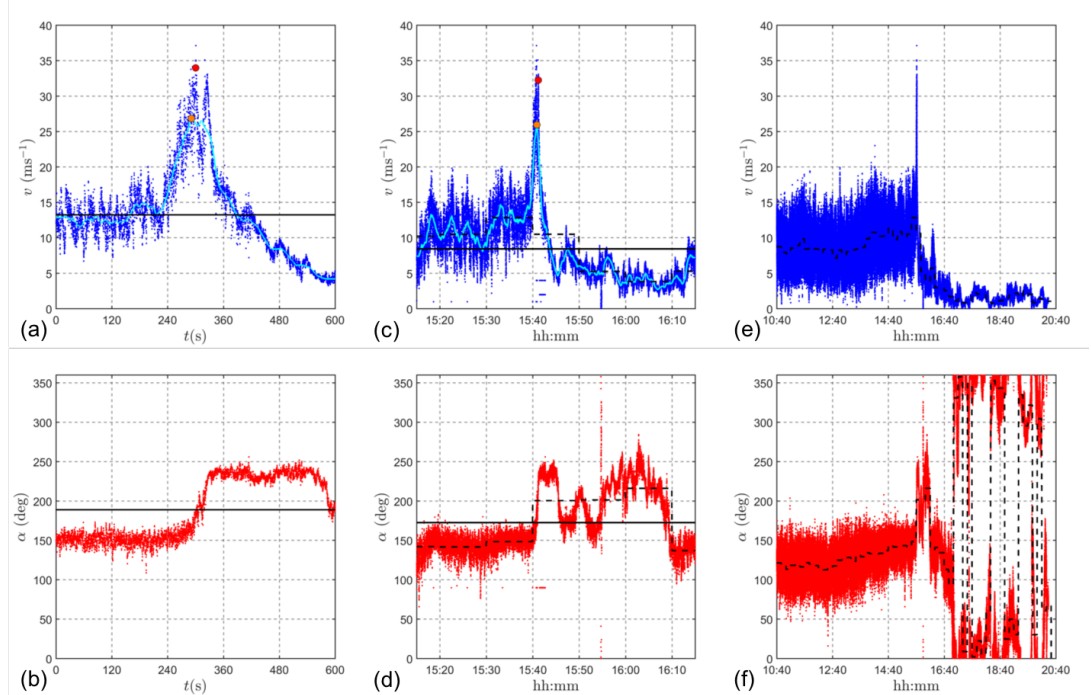

**Figure 8: Thunderstorm outflow recorded on 25 October 2011 at about 15:40 UTC by the anemometer 3 of the Port of La Spezia: wind speed time-series in 10-min (a), 1-h (c), and 10-h (e); wind direction time-series in 10-min (b), 1-h (d), 10-h (f).**





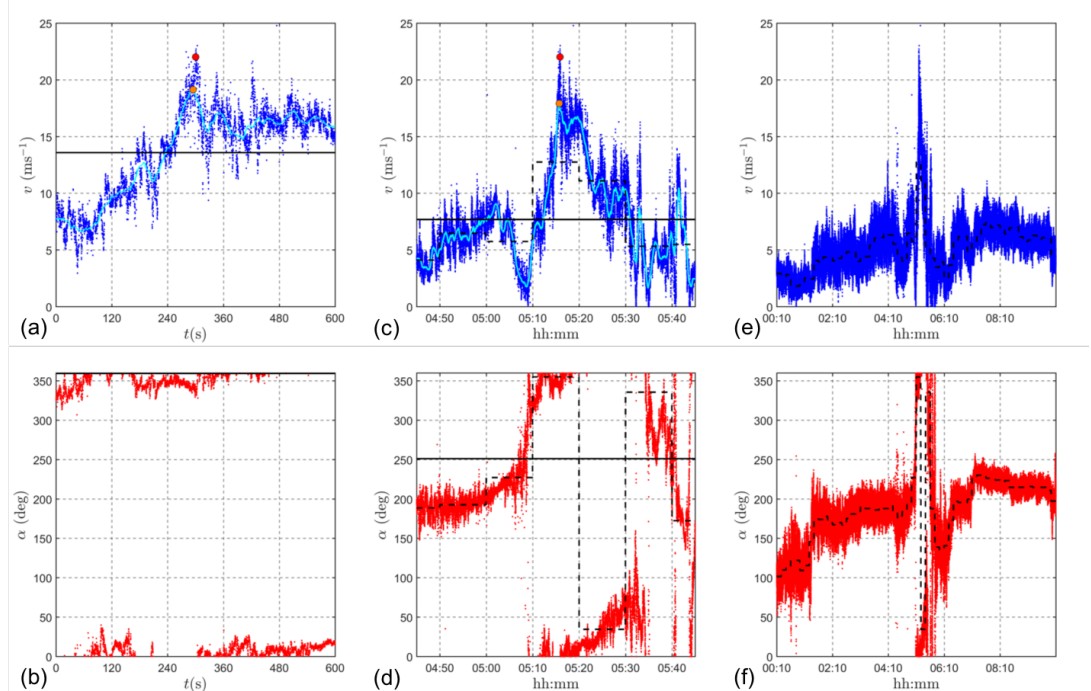

**Figure 9: Thunderstorm outflow recorded on 4 October 2015 at about 05:15 UTC by the anemometer 1 of the Port of Livorno: wind speed time-series in 10-min (a), 1-h (c), and 10-h (e); wind direction time-series in 10-min (b), 1-h (d), and 10-h (f).**





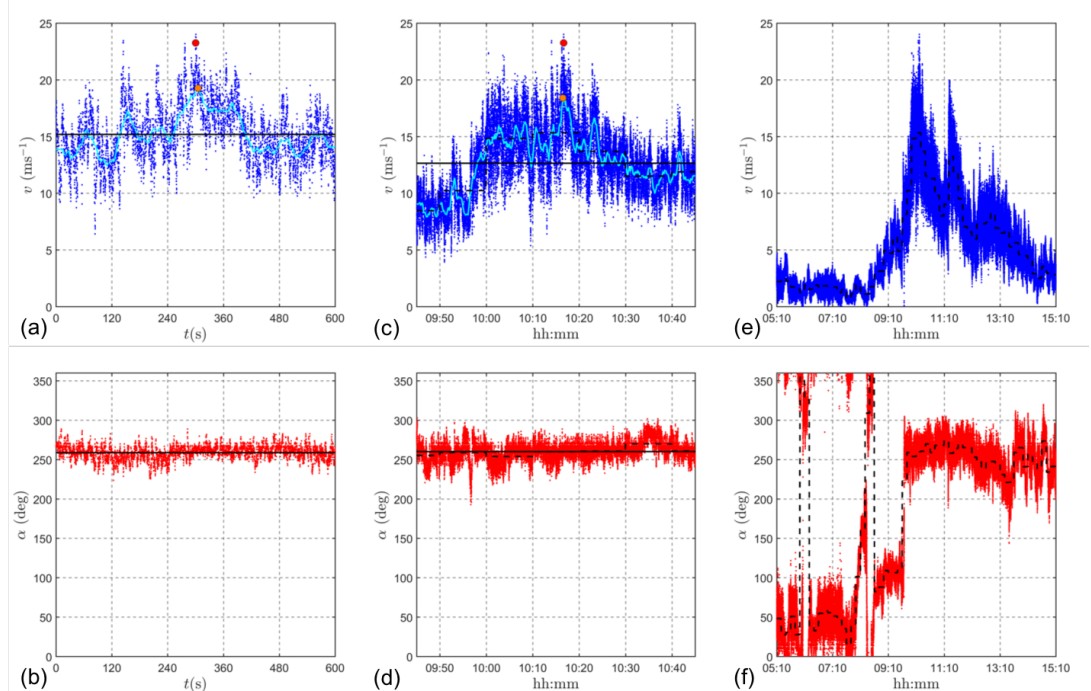

**Figure 10: Thunderstorm outflow recorded on 21 November 2013 at about 10:15 UTC by the anemometer 2 of the Port of Genoa: wind speed time-series in 10-min (a), 1-h (c), and 10-h (e); wind direction time-series in 10-min (b), 1-h (d), and 10-h (f).**



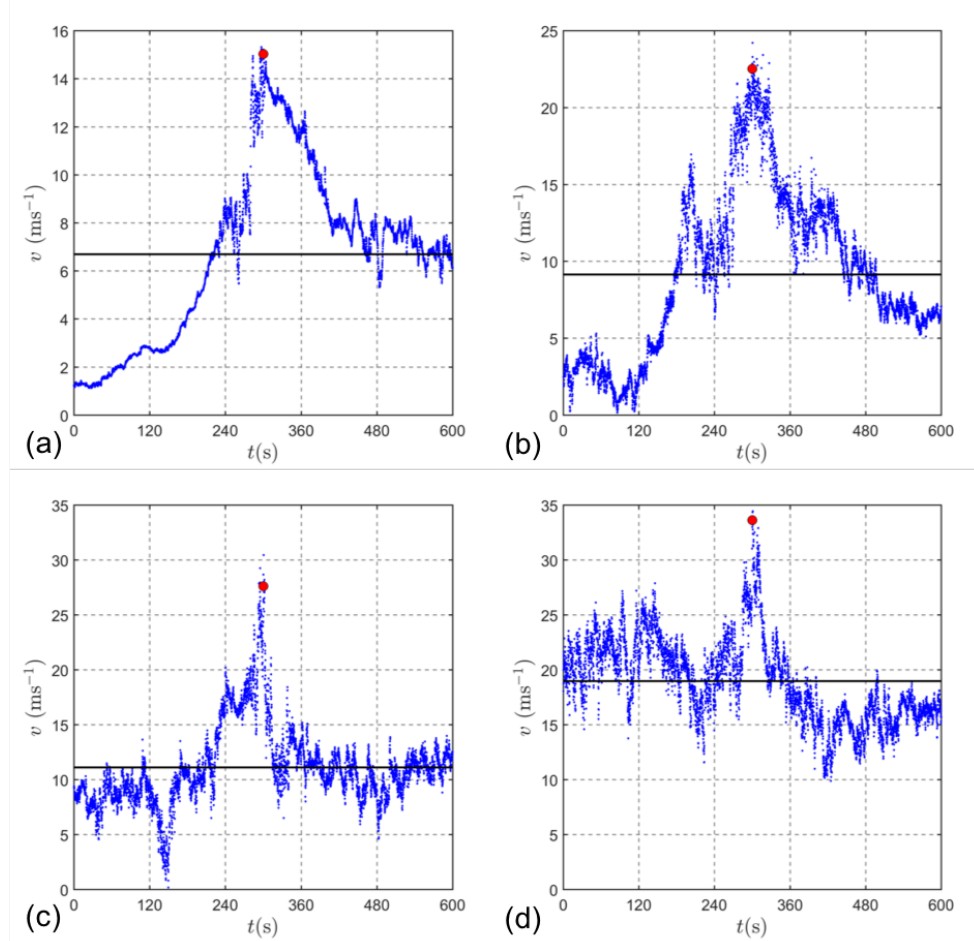

Figure 11: Thunderstorm outflow records characterized by different wind speed classes (a) recorded on 25 August 2015 at 03:20 UTC by the anemometer 1 of the Port of Livorno; (b) recorded on 15 October 2012 at 00:20 UTC by the anemometer 3 of the Port of La Spezia; (c) recorded on 16 November 2010 at 02:00 UTC by the anemometer 5 of the Port of Livorno; (d) recorded on 16 December 2011 at 22:50 UTC by the anemometer 1 of the Port of Livorno.



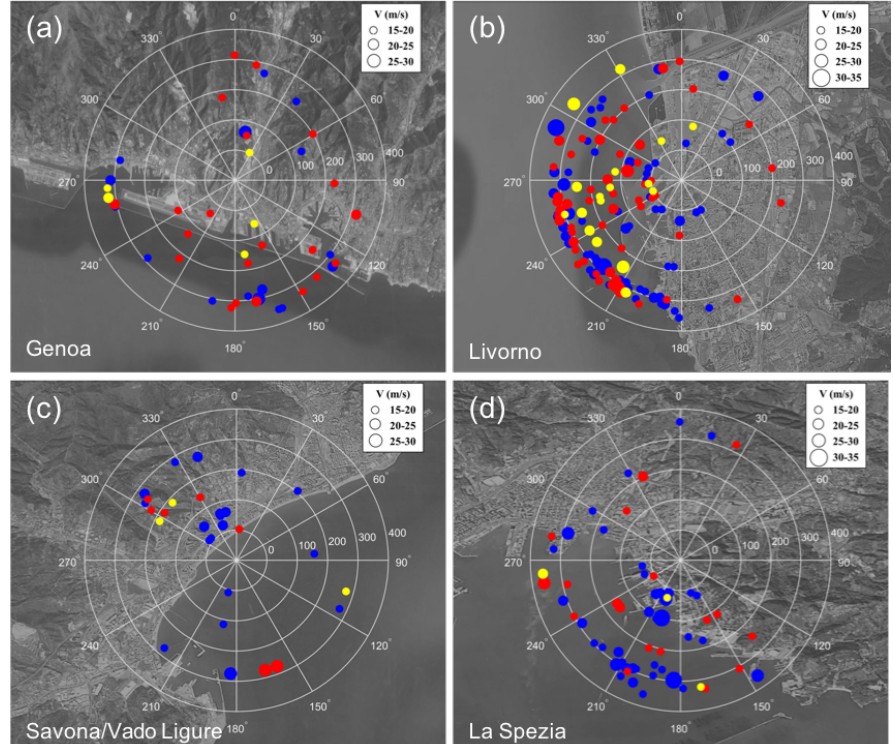

**Figure 12: Day of the year (radial distance from the origin), direction of the thunderstorm outflows and peak wind speed (marker size). Blue triangles: 10-min family; red squares: 1-h family; yellow circles: 10-h family.**





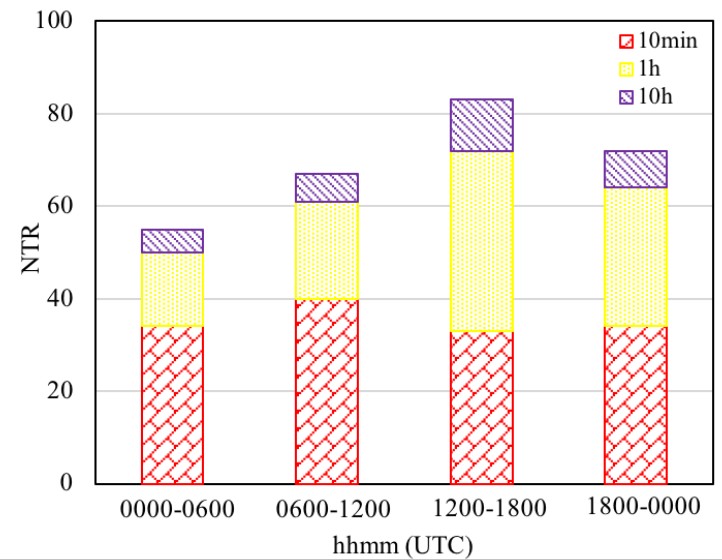

**Figure 13: Number of thunderstorm outflow records detected at different hours of the day.**



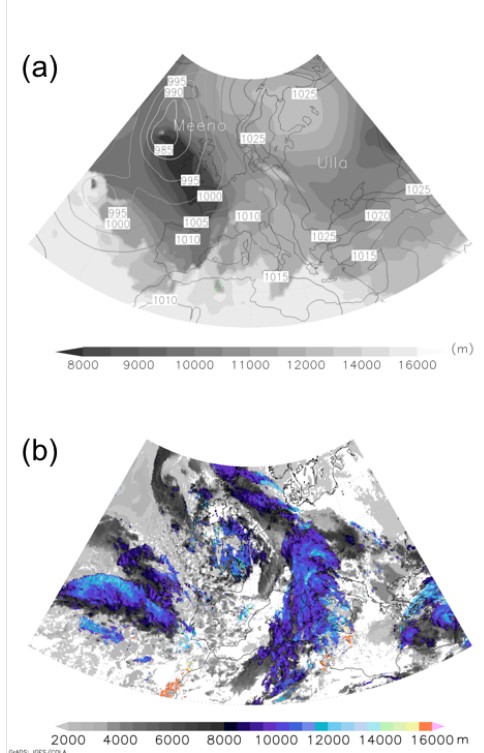

**Figure 14: Top panel (a): mean sea level pressure (contours) and tropopause height (shaded contours) over Europe from GFS analyses on 25 October 2011 at 18:00 UTC. Bottom panel (b): cloud top height from MSG data on 25 October 2011 at 15:45 UTC.**



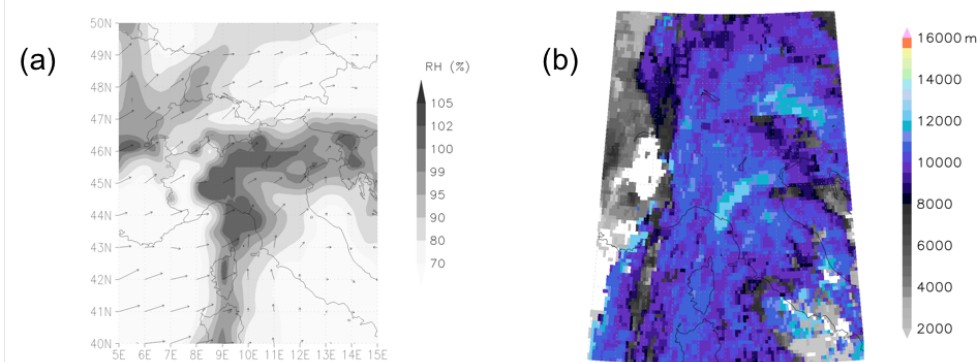

**Figure 15: Panel (a): relative humidity (shaded contours) at 700 hPa and mean storm motion (vectors) from GFS analyses on 25 October 2011 at 18:00 UTC. Panels (b): cloud top height from MSG data on 25 October 2011 at 15:45 UTC.**





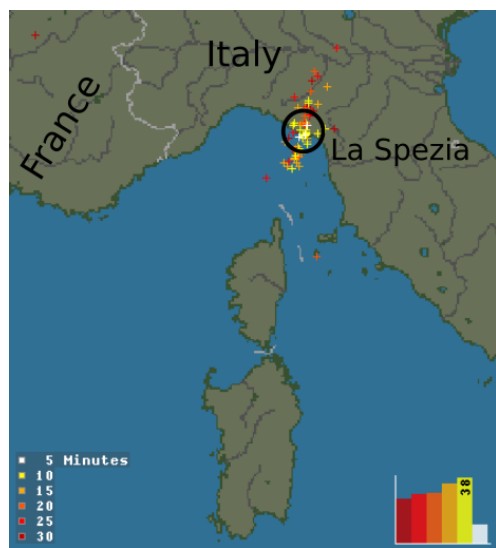

**Figure 16: Strikes recorded on 25 October 2011, from 15:25 to 15:55 UTC, by means of the Blitzortung network for lightning and thunderstorms, retrieved through the online archive. Courtesy Blitzortung.org.**



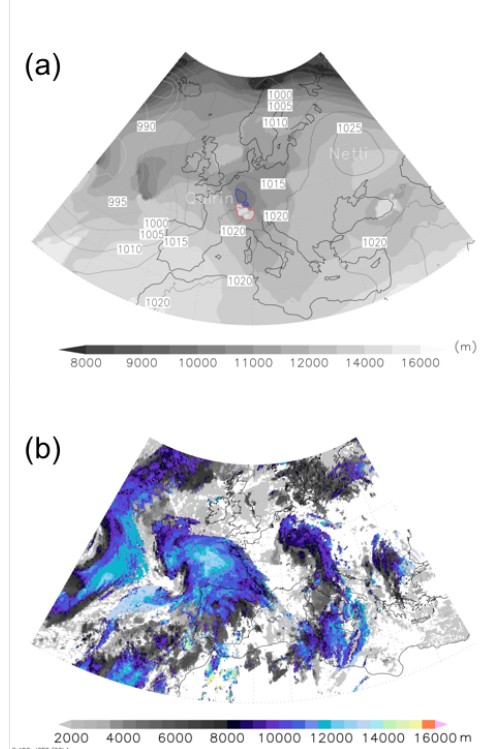

**Figure 17: Top panel (a): mean sea level pressure (contours) and tropopause height (shaded contours) over Europe from GFS analyses. The red (blue) contour corresponds to the tropopause height equal to 12000 m (10000 m) to the north of the Alps. Bottom panel (b): cloud top height from MSG data. Both panels correspond to 4 October 2015 at 06:00 UTC.**



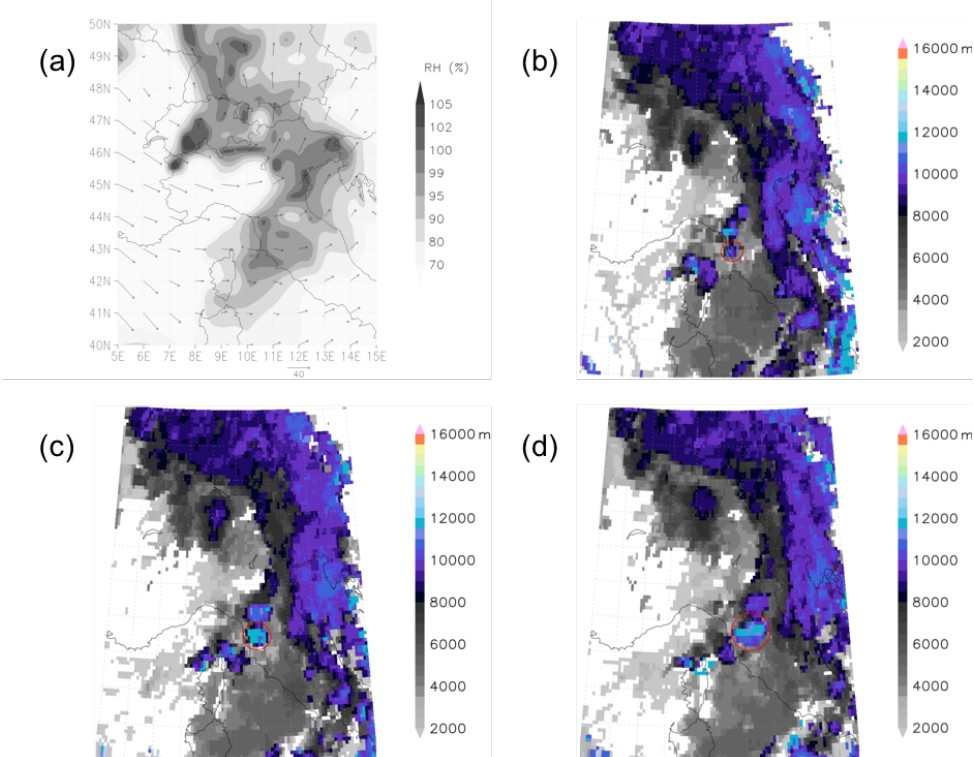

**Figure 18: Panel (a): relative humidity (shaded contours) at 700 hPa and mean storm motion (vectors) from GFS analyses on 4 October 2015 at 06:00 UTC. Panels (b-d): cloud top height from MSG data on 4 October 2015 at 04:30 (b), 05:15 (c), and 06:00 UTC (d). Red circles indicate the thunderstorm position and extension.**





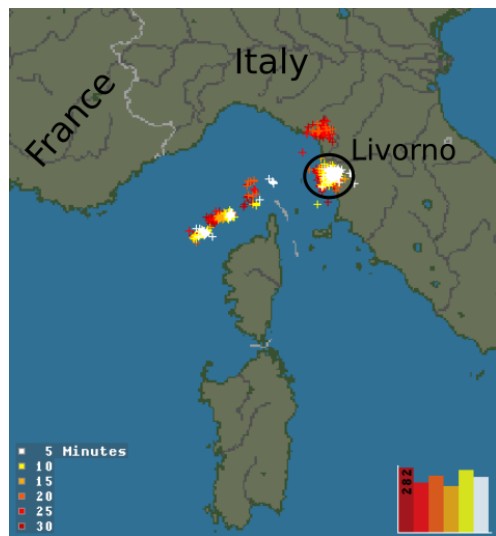

**Figure 19: Strikes recorded on 4 October 2015, from 05:00 to 05:30 UTC, by means of the Blitzortung network for lightning and thunderstorms, retrieved through the online archive. Courtesy Blitzortung.org.**



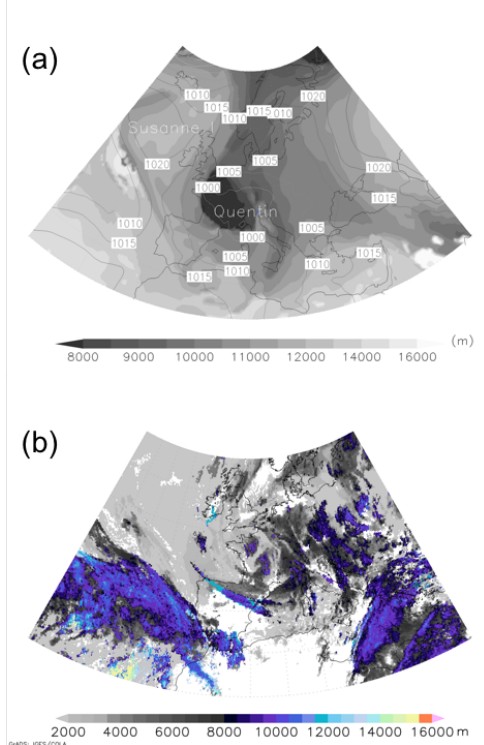

**Figure 20: Top panel (a): mean sea level pressure (contours) and tropopause height (shaded contours) over Europe from GFS analyses on 21 November 2013 at 12:00 UTC. Bottom panel (b): cloud top height from MSG data on 21 November 2013 at 10:15 UTC.**

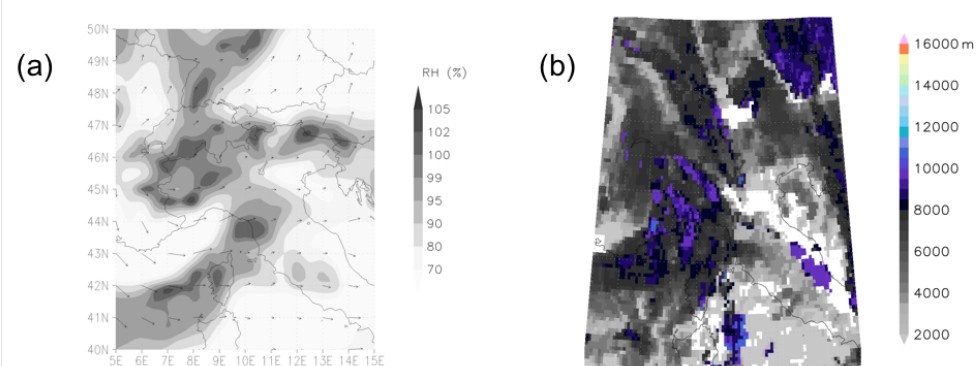

**Figure 21: Panel (a): relative humidity (shaded contours) at 700 hPa and mean storm motion (vectors) from GFS analyses on 21 November 2013 at 1200 UTC. Panels (b): cloud top height from MSG data on 21 November 2013 at 10:15 UTC.**