# Peer review of "Monitoring, cataloguing and weather scenarios of thunderstorm outflows in the Northern Mediterranean"

_Natural Hazards and Earth System Sciences, 2018_

## Referee Comment (RC1) · Anonymous Referee #1 · 3 Apr 2018

Overview

This study analyzes the three events in the Northern Mediterranean from wind engineering and atmospheric sciences points of view. The manuscript is nicely organized and easy to follow. This reviewer believes that this study brings new insights into the nature of downburst events and helps at bridging the gap between meteorology and wind engineering. Several minor comments and suggestions for improvement are enclosed below.

Recommendation: Minor Revision.

Comments 1. Page 8, Paragraph of Line 11. During night, sea surface is warmer than

land due to the higher thermal capacity of water. Therefore, the land-to-sea breezes bring cold air from land to sea and the advected air becomes statically unstable thus resulting in convection. That convection, if strong enough, should be one of the main contributors to the development of thunderstorms over sea surface. Please try to include this description in one or another form.

2. Please provide sources (credit) for Figure 1.

3. Since all meteorological data that are used for the analysis of synoptic and mesoscale conditions are freely available online, I believe it should be possible to make a code (a Python or Matlab script) that would access the websites (of ftp servers) which contain meteorological data and download the desired data automatically for the investigated event. That is, let's say the thunderstorm database contains 100 downburst events that needs to be analyzed and let's say that the important meteorological conditions for a downburst event are A, B, C, D, E, and F (whatever these might be). Then the hypothetical script would access the reanalysis data, satellite data, lightning data, etc. and automatically extract A, B, C, D, E, and F, and further process them. This reviewer believes that such database would be beneficial and it could be as automatic as the procedure developed by De Gaetano et al. (2014) for separation of wind records into three families. Maybe the authors could comment on this subject in the manuscript.

---

## Referee Comment (RC2) · Anonymous Referee #2 · 12 Jul 2018

This paper presents new promising insights about downburst events. It is clearly organized and pleasant reading, therefore highly recommendable to its publication. However, some minor comments and suggestion are done to improve and clarify mostly the introduction section. These comments are listed below.

Suggestion: minor changes

Page 1:

Line 18: Please change to read "10-min", "1-h" and "10-h".

Line 25: Please clarify to which it refers with "Their".

[Figure]

Line 26: Change to read "A climatological condition. . .".

Line 33: Change to read "These are synoptic. . .".

Line 35: Please clarify to which it refers with "Their".

Line 37: Please clarify to which it refers with "After over half century. . .".

Page 2:

Line 4: Please consider adding information about that not all the thunderstorms produce intense radial outflows.

Line 20: Change to read: "However, despite this huge amount of research, this matter. . .".

Line 25: It is recommended adding some information about what an Aeolian event is.

Page 3:

Line 6: It is recommended removing "however" from the sentence. It's redundant.

Line 9: Change to read ". . .the City of Livorno, Italy, was selected. . ."

Line 11: Change to read ". . .all the meteorological data available in this area, which included model analyses, standard in-situ measurements. . .".

Line 17 to 21: Please split the sentence in different ones, is too long.

Line 38 to 41: Please split the sentence in different ones and clarify.

Page 4:

Line 16: Change to read ". . .of 31 anemometers that the current WP and WPS network is made up of, with h being their height. . ."

Page 5:

Line 35: The 1-s peak wind velocity should be also visible in the wind direction? Please

clarify.

Page 8:

Line 19: Change to read ". . . to develop a faster approach that integrates the data. . ."

Page 20:

Figure 1: Please provide credit.

Page 22:

Figure: 3: Please provide credit.

---

## Author Comment (AC1) · 23 Jul 2018

Overview

This study analyzes the three events in the Northern Mediterranean from wind engi- neering and atmospheric sciences points of view. The manuscript is nicely organized and easy to follow. This reviewer believes that this study brings new insights into the nature of downburst events and helps at bridging the gap between meteorology and wind engineering. Several minor comments and suggestions for improvement are enclosed below.

Thank you for your positive comments. All the required changes have been implemented in the text or commented otherwise.

Recommendation: Minor Revision.

Comments 1. Page 8, Paragraph of Line 11. During night, sea surface is warmer than land due to the higher thermal capacity of water. Therefore, the land-to-sea breezes bring cold air from land to sea and the advected air becomes statically unstable thus resulting in convection. That convection, if strong enough, should be one of the main contributors to the development of thunderstorms over sea surface. Please try to include this description in one or another form.

Thank you for your suggestion. The phenomenon that you mentioned has been included in the text.

2. Please provide sources (credit) for Figure 1.

Done. Picture (a) has been changed with another satellite image.

3. Since all meteorological data that are used for the analysis of synoptic and mesoscale conditions are freely available online, I believe it should be possible to make a code (a Python or Matlab script) that would access the websites (of ftp servers) which contain meteorological data and download the desired data automatically for the investigated event. That is, let's say the thunderstorm database contains 100 downburst events that needs to be analyzed and let's say that the important meteorological conditions for a downburst event are A, B, C, D, E, and F (whatever these might be). Then the hypothetical script would access the reanalysis data, satellite data, lightning data, etc. and automatically extract A, B, C, D, E, and F, and further process them. This reviewer believes that such database would be beneficial and it could be as automatic as the procedure developed by De Gaetano et al. (2014) for separation of wind records into three families. Maybe the authors could comment on this subject in the manuscript.

Thanks for your comment. We have included it as a further perspective of development of the present research in the end of Section "Conclusions and perspectives".

---

## Author Comment (AC2) · 23 Jul 2018

This paper presents new promising insights about downburst events. It is clearly organized and pleasant reading, therefore highly recommendable to its publication. However, some minor comments and suggestion are done to improve and clarify mostly the introduction section. These comments are listed below.

Thank you for your positive comments. All the required changes have been implemented in the text or commented otherwise.

Suggestion: minor changes

Page 1:

Line 18: Please change to read "10-min", "1-h" and "10-h".

Done.

Line 25: Please clarify to which it refers with "Their".

The sentence has been changed to make it clearer.

Line 26: Change to read "A climatological condition. . .".

Done.

Line 33: Change to read "These are synoptic. . .".

Done.

Line 35: Please clarify to which it refers with "Their".

The sentence has been changed to make it clearer.

Line 37: Please clarify to which it refers with "After over half century. . .".

The sentence has been changed to make it clearer.

Page 2:

Line 4: Please consider adding information about that not all the thunderstorms produce intense radial outflows.

The sentence has been changed to clarify the aspect that strong downburst are possible but they are not that common.

Line 20: Change to read: "However, despite this huge amount of research, this matter. . .".

Done.

Line 25: It is recommended adding some information about what an Aeolian event is.

We have substituted the term "Aeolian" with the more general term "meteorological".

Page 3:

Line 6: It is recommended removing "however" from the sentence. It's redundant.

Done.

Line 9: Change to read ". . .the City of Livorno, Italy, was selected. . ."

Done.

Line 11: Change to read ". . .all the meteorological data available in this area, which included model analyses, standard in-situ measurements. . .".

Done.

Line 17 to 21: Please split the sentence in different ones, is too long.

Done.

Line 38 to 41: Please split the sentence in different ones and clarify.

We have partly rewritten this sentence and made it shorter. A new reference to Mohr et al. (2017) has been also added.

Page 4:

Line 16: Change to read ". . .of 31 anemometers that the current WP and WPS network is made up of, with h being their height. . ."

Done.

Page 5:

Line 35: The 1-s peak wind velocity should be also visible in the wind direction? Please clarify.

This sentence has been clarified.

Page 8:

Line 19: Change to read ". . . to develop a faster approach that integrates the data. . ."

Corrected, thank you.

Page 20:

Figure 1: Please provide credit.

Done. Picture (a) has been changed with another satellite image.

Page 22:

Figure: 3: Please provide credit.

Done.